# Developing an assistive technology usability questionnaire for people with neurological diseases

**Maria Masbernat-Almenara**[1,2,3], **Francesc Rubi-Carnacea**[1,2,3]*, **Eloy Opisso**[4,5,6], **Esther Duarte-Oller**[7,8], **Josep Medina-Casanovas**[4,5,6], **Fran Valenzuela-Pascual**[1,2,3]

1 Department of Nursing and Physiotherapy, University of Lleida, Lleida, Spain, 2 Research Group of Health Care, IRB Lleida, Institute for Biomedical Research Dr. Pifarré Foundation, Lleida, Spain, 3 Group on Society Studies, Health, Education and Culture, University of Lleida, Lleida, Spain, 4 Institut Guttmann, Neurorehabilitation Institute, Badalona, Spain, 5 Universitat Autònoma de Barcelona, Cerdanyola del Vallés, Barcelona, Spain, 6 Fundació Institut d'investigació en Ciències de la Salut Germans Trias i Pujol, Badalona, Barcelona, Spain, 7 Physical Medicine and Rehabilitation Department, Parc de Salut Mar (Hospital del Mar, Hospital de l'Esperança), Barcelona, Catalonia, Spain, 8 Rehabilitation Research Group, Hospital del Mar Medical Research Institute (IMIM), Barcelona, Catalonia, Spain

* francesc.rubi@udl.cat

**Data Availability Statement:** All relevant data are within the paper and its Supporting Information files.

## Abstract

### Purpose

This study describes the development of a questionnaire for assessing the usability of assistive technologies accessible to people with neurological diseases.

### Methods

A Delphi study was conducted to identify relevant items for the questionnaire. After that, the content validity was addressed to identify the essential items. Once the questionnaire was designed following the results of the Delphi study and content validity, the reliability, validity, and the Rasch model of the questionnaire were examined.

### Results

Two rounds of the Delphi study were carried out. A total of 73 participants (42 experts and 31 users) participated in round 1, and 59 people (27 experts and 32 users) in round 2. A total of 53 and 29 items were identified in rounds 1 and 2, respectively. In the content validity, we found nine items above the threshold of 0.58. Finally, ten items were included in the questionnaire. Fifty-one participants participate in the reliability and validity of the questionnaire. The internal consistency reliability of the questionnaire analyzed by Cronbach's Alpha was α = 0,895. There was moderate to considerable concordance among our questionnaire items test-retest in the Kappa coefficient and a strong association between test-retest in the Spearman's coefficient ρ = 0.818 (p<0,001). The intraclass correlation coefficient was 0,869 with a 95% confidence interval (0,781;0,923). There was a strong correlation between the total scores of the new questionnaire and other validated questionnaires analyzed with Spearman's coefficient ρ = 0.756 (p<0,001). The ten items demonstrated a satisfactory fit to the Rasch model.

**Funding:** The funders had no role in study design, data collection and analysis, decision to publish, or preparation of the manuscript.

**Competing interests:** The authors have declared that no competing interests exist.

## Conclusions

The present study suggested that the new questionnaire is a reliable 10-item usability questionnaire that allows subjective and quick assessment of the usability of assistive technologies by people with neurological diseases.

## Introduction

Neurological disorders are among the most important causes of disability (247–308 million) and death (8·8–9·4 million) worldwide [1]. In addition, the burden of neurological disorders in public health has increased substantially in the last 25 years because of expanding population numbers, aging, and increased survivor rates from stroke and other neurological disorders [2]. These survivors require intensive rehabilitation to reduce the sequelae of the disorders, increase their quality of life, and improve their autonomy in activities of daily living. In most cases, some assistive technology is needed.

Recent advances in technology and rehabilitation have led to the development of new tools to assess people with disabilities and improve their functioning and autonomy in their daily life [3]. It is known that a good acceptance of assistive technology can improve the quality of life and social inclusion of these patients [4]. However, not all products achieve the goal for which they were designed since they often do not consider the real needs of users [5]. Therefore, it is increasingly necessary to involve end-users from the beginning of developing new products [6]. A product can only be considered successful if it is used, and currently, more than 50% of users abandon their new products because they are not sufficiently usable [7, 8]. For this reason, usability is becoming more critical in engineering and rehabilitation. According to the ISO 9241–11 [9], usability is the extent to which specified users can use a product to achieve specified goals with effectiveness, efficiency, and satisfaction in a specified context of use. Understanding this, the design of a product should not involve only multidisciplinary theoretical foundations, but user experience should be considered throughout the design process to ensure quality, improve usability, and increase product acceptability [5, 10]. For that reason, it is essential to perform a usability test during product development to know if it fits the user's needs. Usability testing refers to evaluating a product or service by testing it with representative users [11].

According to the WHO, assistive technology is a general term covering the systems and services related to delivering assistive products and services [12]. Assistive products aim to maintain or improve an individual's functioning and independence, promoting their well-being [12].

There is a lack of evidence-based procedures for assistive technologies selection [13–15]. For example, professionals prescribe assistive products without considering the user's needs. However, this clinical outcome assessment is important in clinical practice and research because it improves evidence and provides considerable feedback to healthcare professionals and patients, enhancing their empowerment with their opinions and needs and improving their quality of life [16].

Many existing usability questionnaires have been developed to evaluate software usability and web accessibility [17] and not assistive technologies [18]. Some questionnaires [19–21] measure the psychosocial impact of quality of life by using assistive technologies from the point of view of people with disabilities. Those are interesting questionnaires; however, they do not evaluate the usability of assistive technologies per se. There are other questionnaires

developed exclusively to assess wheelchairs [22], and only a few are explicitly developed for assistive technologies. For example, the Quebec User Evaluation of Satisfaction with Assistive Technology (QUEST 2.0) [23] contains technical items such as weight, product dimensions, service delivery, repairs, and device services. People with neurological diseases have limitations in the accessibility of the existing usability questionnaires, for example, in the comprehension of the questions or the answer form. In addition, existing questionnaires are too extensive for them or do not address all the essential items for assessing usability and user satisfaction [18, 24]. It is known that longer questionnaires incur high costs in data collection and reduce the number of answers due to the time answering and the quality of the information gathered due to fatigue [25, 26].

The objective of this study was to develop a short questionnaire focused only on the usability of assistive technology products and user satisfaction, and it should be accessible and easy to understand to people with neurological diseases.

## Materials and methods

### Questionnaire design

**Delphi study.** To construct an easy-to-use questionnaire, we performed a Delphi study [27], as it is a prospective approach that seeks to derive a consensus from a group of experts based on the analysis and reflection of a defined problem. A google form questionnaire was sent via email to the neurological healthcare professionals from the Institut Guttmann, Spain and Hospital de l'Esperança, Spain. Also, all the patients from the Institut Guttmann who voluntarily agreed to participate were interviewed to answer the questionnaire.

The questionnaire contains six questions: three were open questions related to assistive technologies, and three were multiple-choice questions about how a questionnaire should be easy and quick to answer. The three open questions were: (1) List some requirements or characteristics important to how the assistive products should or should be manufactured. (2) Would you change anything about the existing assistive products? (3) What kind of assistive products would you like to have? (for users)/ in your workplace? (experts). The three multiple-choice questions related to the questionnaires were: (1) the time required for you to answer a questionnaire, (2) which questionnaire format the user preferred, and (3) how many questions you would like to answer.

**Content validity ratio.** Once the Delphi study finished, and the items were obtained, they were evaluated using the content validity ratio (CVR) [28, 29]. Content validity addresses the degree to which items of an instrument sufficiently represent the content domain and answers the question of to what extent the selected sample in an instrument or instrument items is a comprehensive sample of the content [29]. The content validity ratio varies between 1 and -1. Higher scores indicate greater agreement by panel members on the need for an item in an instrument. The formula of content validity ratio is $CVR = (N_e - N/2)/(N/2)$, in which the $N_e$ is the number of panelists indicating "essential" and N is the total number of panelists [29]. The numeric value of the content validity ratio is determined by Lawshe Table [28].

**Questionnaire design.** The questions were written based on the qualitative data obtained in the Delphi study and the items obtained in the CVR. Subsequently, ten users (people with neurological diseases) read the questionnaire to assess their understanding of the questions and if their answers from the Delphi study could be represented with the proposed response values (numbers, traffic light colors, and faces).

**Questionnaire reliability and validity.** The reliability and validity of the new questionnaire were addressed.

*Sample size*. The recommended sample size for similar studies has been established as 50 subjects [30, 31]. The inclusion criteria for participants were (1) adults (≥18 years old) with neurological diseases that have been using some assistive products for the past month. Exclusion criteria moderate to severe cognitive impairment based on the Pfeiffer Short Portable Mental State Questionnaire (Pfeiffer SPMSQ) [32] translated into Spanish [33].

*Assistive product analyzed*. It is challenging to analyze the same product because neurological patients have personalized assistive technologies adapted to their disability. Therefore, all the participants could choose which product addressed, only with the condition that they had used it for at least one month.

**Data analysis of reliability.** The reliability and validity of the new questionnaire and all the data were analyzed using the SPSS Statistics 27 program.

The internal consistency reliability of the new questionnaire was analyzed using Cronbach's Alpha [34]. The value of the coefficients was interpreted as follows: unacceptable ($0.5 \geq \alpha$), poor ($0.6 \geq \alpha \geq 0.5$), questionable ($0.7 \geq \alpha \geq 0.6$), acceptable ($0.8 \geq \alpha \geq 0.7$). good ($0.9 \geq \alpha \geq 0.8$), and excellent $\alpha \geq 0.9$) [34].

*Test-retest*. Reliability can be assessed with a test-retest comparison. This method evaluates the stability of the results at two different points in time using a stability coefficient. To perform the test-retest, the same questionnaire was administered on two occasions, separated by a certain period, to the same subjects and under the same conditions. According to different authors [35, 36], an interval between two days and two weeks between the test-retest interviews is recommended. Accordingly, the interval between the two tests was set to 15 days. In order to know the concordance between the test and the retest, data was analyzed by weighted quadratic Kappa coefficient [37]. The value of the coefficients was interpreted as follows: poor ($< 0.00$), weak (0.00–0.20), good (0.20–0.40), moderate (0.41–0.60), considerable (0.61–0.80), and almost perfect (0.81–1.00) [37].

Intraclass correlation coefficient (ICC) model 1.1 with a 95% of confidence interval (CI) was measured. ICC indicates the degree to which the participants maintained their opinion during the test-retest [38]. The value of the coefficients was interpreted as follows: poor reliability ($< 0.5$), moderate reliability (0.5–0.75), good reliability (0.75–0.9), and excellent reliability ($> 0.90$) [38].

The normality of the data was tested using the Kolmogorov-Smirnov test, and the repeatability was determined by the Spearman's correlation coefficient [39] between the test and the retest.

## Data analysis of validity

*Concurrent validity*. To know the relationship between the new questionnaire and another related questionnaire, QUEST 2.0 [23] was chosen. It is the shortest worldwide questionnaire and has been translated and validated in several languages. Only eight items of the QUEST 2.0 was administered to users in Spanish [40] because the other four items are related to services. Also, in this study, data was tested using the Kolmogorov-Smirnov test and analyzed using Spearman's correlation coefficient [39].

*Measurement of other parameters*. The time was measured while the participants answered the questionnaire to obtain the average time of all the participants.

**Rasch model.** In order to further analyze the construct validity of the different items, the Rasch model [41] was carried out. This model gives an idea of the scale's internal consistency by relating the item's difficulty to the person's ability [41]. The values of infit and outfit mean squares (MNSQ) of 1 indicate a perfect fit between the data and the model, values between 0.5 and 1.5 indicate an acceptable fit, and values greater than 2 indicate a severe mismatch [41].

The study was developed following the COSMIN guidelines [42].

### Ethical approval

The Institut Guttmann Neurorehabilitation Hospital Ethics Committee approved this study. In addition, this research was conducted following the Declaration of Helsinki's ethical principles. All participants participated voluntarily, and they signed an informed written consent form. Their personal data were archived following the Spanish Organic Law 3/2018, December 5, on the Protection of Personal Data and guarantee of digital rights.

## Results

### Study Delphi

Two rounds of a Delphi study were needed to obtain all the items of the new questionnaire. A total of 73 participants (42 experts and 31 users) were involved in round 1 (Table 1). From round 1, 53 different items and qualitative information about questionnaires were derived qualitatively.

Round two involved 59 people (27 experts and 32 users) (Table 1). This round obtained 15 items and data about the scale. The items were: "effectiveness", "comfort", "adaptability", "easy to put on/off", "safe", "lightweight", "functioning", "ergonomic", "economical", "affordable", "easy to use", "feedback", "stimulating," "monitored" and "movement facilitator. Due to "economical" and "affordable" have the same meaning, both were considered as a single item. Additionally, in this round and following the usability premises, the items "aesthetics" and "easy to remember how to use it" were added. Finally, 16 items were analyzed using the content validity ratio.

### Content validity ratio (CVR)

Thirty-four experts from the Delphi study from the Institut Guttmann and the Hospital de l'Esperança were selected to evaluate the items obtained from round two. 70% of the participants have more than ten years of expertise in neurorehabilitation, and all of them used to work with assistive technologies experts (Table 2).

Participants had to choose which items were *essential*, *useful but not essential*, and *not essential* when evaluating the items. Table 3 shows the results. Nine of the sixteen items exceeded the threshold of 0.58. In addition, the experts agreed to accept the item "comfortable" because its threshold was 0.53.

The experts considered "functional" and "movement facilitator" as one word due to the similarity of their meanings. The item "satisfaction" was added to the questionnaire to ascertain the end-user opinion of the product. Therefore, ten items were selected to create the questionnaire (Table 4).

### Questionnaire design

Following the qualitative information from the Delphi study, the questionnaire was formulated in an understandable language, and the length was as short as possible. When necessary, the users could fill the blank space with the product's name being evaluated. The 6-point Likert scale was chosen since it forces the respondent to decide positively or negatively according to the item in question [43]. A numeric panel from 0 to 5 points was used, and each number was associated with a box using the colors of the traffic light to facilitate the choice. Furthermore, faces with expressions were added to facilitate the answers according to the patients with neurological diseases responses from the Delphi study.

The statements of the questions are in first person to facilitate users' answers and usability experiences of the product subjectively [44].

**Table 1. Demographic characteristics of participants of the two rounds in the Delphi study.**

| EXPERT PANEL | Round 1 n = 42 | | Round 2 n = 27[a] | |
|---|---|---|---|---|
| Demographic Characteristics | Number | Percentage | Number | Percentage |
| **Age** | | | | |
| Between 18 and 25 years | n = 3 | 7% | n = 1 | 4% |
| Between 26 and 35 years | n = 17 | 41% | n = 11 | 41% |
| Between 36 and 45 years | n = 11 | 26% | n = 8 | 29% |
| Between 46 and 65 years | n = 11 | 26% | n = 7 | 26% |
| More than 65 years | n = 0 | 0% | n = 0 | 0% |
| **Sex** | | | | |
| Female | N = 30 | 71% | 19 | 70% |
| Male | N = 12 | 29% | 8 | 30% |
| **Profession** | | | | |
| Physician | n = 9 | 21% | 6 | 22% |
| Physiotherapist | n = 14 | 33% | 9 | 33% |
| Occupational Therapist | n = 13 | 31% | 8 | 30% |
| Other[b] | n = 6 | 15% | 4 | 15% |
| **Expert in Neurorehabilitation** | | | | |
| Yes | 39 | 93% | 27 | 100% |
| USER PANEL | Round 1 n = 31 | | Round 2 n = 32[d] | |
| Demographic Characteristics | Number | Percentage | Number | Percentage |
| **Age** | | | | |
| Between 18 and 25 years | n = 4 | 13% | n = 1 | 3% |
| Between 26 and 35 years | n = 6 | 19% | n = 9 | 28% |
| Between 36 and 45 years | n = 5 | 16% | n = 3 | 10% |
| Between 46 and 65 years | n = 13 | 42% | n = 16 | 50% |
| More than 65 years | n = 3 | 10% | n = 3 | 9% |
| **Sex** | | | | |
| Female | n = 17 | 45% | n = 16 | 50% |
| Male | n = 14 | 55% | n = 16 | 50% |
| **Pathology** | | | | |
| Spinal Cord Injury | n = 13 | 42% | n = 7 | 22% |
| Traumatic Brain Injury | n = 5 | 16% | n = 1 | 3% |
| Stroke | n = 8 | 26% | n = 13 | 41% |
| Other | n = 5 | 16% | n = 11 | 34% |
| **Assistive Technologies** | | | | |
| Wheelchair | n = 23 | 53% | n = 19 | 36% |
| Canes | n = 8 | 18% | n = 12 | 23% |
| Orthesis | n = 8 | 19% | n = 15 | 28% |
| Other | n = 2 | 5% | n = 6 | 11% |
| Nothing[c] | n = 2 | 5% | n = 1 | 2% |

[a]In the second round, 36% did not answer the questionnaire.

[b]Others are, for example, orthopedic professionals and engineers.

[c]Nothing means that he/she did not use assistive technologies at the moment of the interview; however, she/he used them in the past.

[d]Users are different from round 1.

Once the questionnaire was finished, ten users (see demographic characteristics in Table 5) read and answered the questionnaire to know if all the requirements from the Delphi study were met. During this process, some wording modifications were made.

**Table 2. Demographic characteristics of participants in content validity.**

| Demographic Characteristics | Content Validity n = 34 | |
|---|---|---|
| | **Number** | **Percentage** |
| **Age** | | |
| Between 26 and 35 years | n = 8 | 24% |
| Between 36 and 45 years | n = 12 | 35% |
| Between 46 and 65 years | n = 14 | 41% |
| **Sex** | | |
| Female | N = 21 | 62% |
| Male | N = 13 | 38% |
| **Profession** | | |
| Physician | n = 11 | 32% |
| Physiotherapist | n = 14 | 41% |
| Occupational Therapist | n = 7 | 21% |
| Other[a] | n = 2 | 6% |
| **Expert in Neurorehabilitation[b]** | | |
| Between 1 and 5 years | n = 5 | 15% |
| Between 5 and 10 years | n = 5 | 15% |
| Between 10 and 20 years | n = 12 | 35% |
| More than 20 years | n = 12 | 35% |

[a]Others are, for example, orthopedic professionals and engineers.

[b]All the participants used to work with people with neurological diseases.

The new questionnaire was named "Assistive Technology Usability Questionnaire for people with Neurological diseases" (NATU Quest) and included ten questions. It should be administered at the end of a usability test. The final version of the questionnaire, questionnaire score, and interpretation are available in the supporting information file.

**Table 3. Content validity ratio results.**

| Items | ne | N | CVR[a] |
|---|---|---|---|
| 1. Effectiveness | 33 | 34 | 0,97 |
| 2. Comfortable | 18 | 34 | 0,53 |
| 3. Adaptability | 21 | 34 | 0,62 |
| 4. Easy to put on/off | 20 | 34 | 0,59 |
| 5. Safe | 32 | 34 | 0,94 |
| 6. Lightweight | 12 | 34 | 0,35 |
| 7. Functionality | 31 | 34 | 0,91 |
| 8. Ergonomic | 22 | 34 | 0,65 |
| 9. Economical/ Affordable | 13 | 34 | 0,38 |
| 10. Easy to use | 24 | 34 | 0,71 |
| 11. Aesthetics | 4 | 34 | 0,12 |
| 12. Easy to remember how to use it | 20 | 34 | 0,59 |
| 13. Feedback | 16 | 34 | 0,47 |
| 14. Stimulating | 14 | 34 | 0,41 |
| 15. Monitored | 10 | 34 | 0,29 |
| 16. Movement Facilitator | 21 | 34 | 0,62 |

[a]CVR (content validity ratio) = (Ne-N/2) / (N/2) with 34 people in the expert panel (N = 34), the items with a CVR bigger than 0.58 (in bold) will remain at the instrument, and the rest will be eliminated.

**Table 4. Questions and the items that are involved in each question.**

| Questions | | Items |
|---|---|---|
| 1 | I believe that _____ can help me improve my functional independence | Effectiveness |
| 2 | I feel comfortable wearing/using ____ | Comfortable |
| 3 | _____ adapts to my characteristics and needs | Adaptability |
| 4 | Donning/Doffing . . . . . . . . . . . . . . . .. is quick and easy for me. | Easy to put on / off |
| 5 | I feel safe using/wearing _____ / _____ is safe in its use | Safe |
| 6 | _____ allows me to achieve my goal/ allows me to perform a movement/action that I could not do before | Functionality/ movement facilitator |
| 7 | _____ adapts to my special needs. | Ergonomics |
| 8 | In general, _____ is easy to use | Easy to use |
| 9 | Information and instructions of use_____ are easy to understand and easy to remember | Easy to remember how to use it |
| 10 | Overall, I am satisfied with_____ | Satisfaction |

## Reliability and validity

### Sample description

A total of 51 people with neurological diseases consecutive recruited from the Institut Gutt-mann Hospital voluntarily agreed to participate in the study. These people were different from the Delphi study, and their demographic characteristics are summarized in Table 6. Fifty-three

**Table 5. Demographic characteristics users from the questionnaire design.**

| Questionnaire analysis | n = 10 | |
|---|---|---|
| Demographic Characteristics | Number | Percentage |
| **Age** | | |
| Between 18 and 25 years | n = 1 | 10% |
| Between 26 and 35 years | n = 2 | 20% |
| Between 36 and 45 years | n = 2 | 20% |
| Between 46 and 65 years | n = 3 | 30% |
| More than 65 years | n = 2 | 20% |
| **Sex** | | |
| Female | n = 6 | 60% |
| Male | n = 4 | 40% |
| **Pathology** | | |
| Spinal Cord Injury | n = 4 | 40% |
| Traumatic Brain Injury | n = 2 | 20% |
| Stroke | n = 3 | 30% |
| Other | n = 1 | 10% |
| **Assistive Technologies** | | |
| Wheelchair | n = 8 | 80% |
| Canes | n = 1 | 10% |
| Orthesis | n = 4 | 40% |
| Other | n = 1 | 10% |
| Nothing[a] | n = 1 | 10% |

[a]Nothing means that he/she did not use assistive technologies at the moment of the interview; however, she/he used it in the past

**Table 6. Demographic characteristics of the 51 participants in the reliability and validity of NATU quest.**

| Demographic Characteristics | Mean /Number | Percentage |
|---|---|---|
| Age, range 16–72 years | Mean = 48 | |
| **Sex** | | |
| Female | n = 20 | 39% |
| Male | n = 31 | 61% |
| **Pathology** | | |
| Spinal cord Injury | n = 22 | 43% |
| Traumatic Brain Injury | n = 3 | 6% |
| Stroke | n = 16 | 31% |
| Acquired Brain injury | n = 4 | 8% |
| Other neurological diseases | n = 6 | 12% |
| **Education** | | |
| Secondary | n = 13 | 25% |
| High School Diploma | n = 1 | 2% |
| Certificate of Professional Standards | n = 15 | 29% |
| Higher Level Education Cycle | n = 7 | 14% |
| University Degree | n = 15 | 30% |
| **Questionnaire form** | | |
| Self-administration | n = 24 | 53% |
| Interview | n = 27 | 47% |
| **Assistive Technologies** | | |
| Wheelchair | n = 29 | |
| Walking stick | n = 8 | |
| Walker | n = 3 | |
| Splint | n = 7 | |
| Treadmill | n = 5 | |

percent of the participants answered the questionnaire through an interview due to physical limitations, and the rest (47%) answered it by themselves. First, the participants answered the new questionnaire and the QUEST 2.0. On average, the participants completed the new questionnaire within 102.40 seconds in the first administration and within 82.08 seconds in the second administration. The QUEST 2.0 was administered once before the NATU Quest, and the participants needed an average of 74 seconds to complete it. Participants scored an assistive product they had used in the last three months. All the patients answered all the items.

## Reliability results

The internal consistency reliability of the NATU Quest was analyzed using the Cronbach's Alpha [34] (α = 0.895). This result can be interpreted as good reliability.

Reliability through test-retest. A retest was performed 15 days after answering the two questionnaires for the first time to assess the reliability of the NATU Quest. Table 7 shows the weighted quadratic Kappa coefficient and Spearman's coefficient results of the NATU Quest. The results showed a moderate to considerable concordance between NATU Quest items test-retest in the Kappa coefficient because all the results were above 0,50. The results also showed a strong association between test-retest in the Spearman's coefficient (ρ = 0.818), significant with p-value < 0.0001. The results of the ICC showed good reliability (ICC = 0.869; CI 95% 0.781 to 0.923).

**Table 7. NATU quest reliability of the test-retest.**

| NATU Quest | | | | |
|---|---|---|---|---|
| **Items** | | **Weighted Kappa** | **95% Confidence Interval** | |
| | | | **Lower Limit** | **Upper Limit** |
| 1 | Effectiveness | 0.727 | 0.535 | 0.918 |
| 2 | Comfortable | 0.733 | 0.645 | 0.821 |
| 3 | Adaptability | 0.585 | 0.434 | 0.735 |
| 4 | Easy to put on/off | 0.631 | 0.399 | 0.863 |
| 5 | Safe | 0.551 | 0.206 | 0.895 |
| 6 | Functionality | 0.685 | 0.541 | 0.828 |
| 7 | Ergonomics | 0.673 | 0.507 | 0.840 |
| 8 | Easy to use | 0.544 | 0.332 | 0.755 |
| 9 | Easy to remember how to use it | 0.624 | 0.298 | 0.950 |
| 10 | Satisfaction | 0.714 | 0.557 | 0.871 |
| **Spearman** | | 0.818[a] | | |

[a]The correlation is significant p< 0.0001 (bilateral).

## Concurrent validity

The correlation of the total scores between NATU Quest and QUEST 2.0 analyzed with the Spearman's coefficient was strong with $\rho = 0.756$ significant, with p-value < 0.0001.

## Rasch model results

All ten items demonstrated a satisfactory fit to the Rasch model, which could be considered productive for measurement (MNSQ infit between 0.64 and 1.43; MNSQ outfit between 0.52 and 1.49).

## Discussion

There is a need for a short and easy questionnaire to properly assess the usability of assistive technologies in people with neurological diseases.

The items included in the questionnaire, the format of the questionnaire, and the answer form were derived through two rounds of a Delphi study [27] based on the opinion of 69 experts (neurorehabilitation professionals, such as occupational therapists and physiotherapists) and 63 users (people with neurological diseases). Finally, we narrowed items down to 10 essential usability items using a content validity ratio. Some of the items are represented in different words in the other usability questionnaires, for example: "Safe," is included in the PIADS [20] and the QUEST 2.0. [23] and the Usability Scale for Assistive Technology for Wheeled Mobility (USAT-WM) [22], while "comfort" appears in PIADS [20] and QUEST 2.0. [23]. The item "easy to use" appears in QUEST 2.0. [23] and USAT-WM) [22].The items "adaptability", "ergonomic" and "satisfaction" are included in the PIADS [20], while "easy to put on/off," and "effectiveness" are included in QUEST 2.0. [23]. The item "functioning" appear in the USAT-WM [22]. Finally, the item "easy to remember how to use it" does not appear in any questionnaire but is a usability attribute [15].

Once the questionnaire form was designed, 51 end-users with neurological diseases participated in the questionnaire validity and reliability. The results suggested that NATU Quest has good reliability and validity and fits in the Rasch model.

In contrast with other questionnaires, the NATU Quest was developed, considering the opinions of professionals and people with neurological diseases. Other relevant aspects of this study are the heterogeneity of the included sample, the wide range of neurological diseases, and the inclusion of different assistive technologies.

In this study, we developed a usability scale to analyze assistive technologies for people with neurological diseases; however, the study had some limitations: (1) Selection bias of the participants since most of them were from the same province. However, other regions have the same experiences [45]. (2) Although all the users who participated in the validation had a neurological disease, the authors chose the Pfeiffer SPMSQ to assess the cognitive problems because it is short and quick to answer. However, Pfeiffer SPMSQ does not accurately assess all possible cognitive deficits, and it is not sensitive enough to detect low or mild cognitive deficits. (3) Different assistive technologies were analyzed due to the people with neurological disease conditions. It would be very interesting to perform another validation with the same product for all users, which may be a good option for developing a new product. (4) For practical reasons we chose that the users have used the product at least for one month, however, probably is not enough time to test a product.(5) Finally, the test-retest reliability was only compared to QUEST 2.0; because we considered that adding other tests for comparison in the same study would have placed a burden on the users.

It would be interesting to measure test-retest reliability with another usability questionnaire for future work. It would also be interesting to verify the new questionnaire's external validity with other population groups, such as older people. Finally, the items should be reviewed after a few years to determine if they are still sensitive enough to assess rapidly evolving assistive technologies. Likewise, it should be interesting to translate the new questionnaire into other languages.

## Conclusion

The present study suggested that the NATU is a reliable 10-item usability questionnaire that allows subjective and quick assessment of the usability of assistive technologies. This questionnaire aims to be accessible to people with neurological diseases and reflects the level of acceptance and satisfaction a patient has with the product being used. In addition, the NATU Quest can also be useful for evaluating products in development through user-centered design since the patient can state an opinion about the product during its development, which will facilitate the development of products for a better fit for patients' needs.

## Supporting information

**S1 File. NATU questionnaire.**
(PDF)

**S2 File. All data.**
(XLSX)

## Acknowledgments

The authors are grateful to all the professionals and participants involved in this study for their contribution.

## Author Contributions

**Conceptualization:** Maria Masbernat-Almenara, Eloy Opisso, Josep Medina-Casanovas.

**Data curation:** Maria Masbernat-Almenara.

**Formal analysis:** Maria Masbernat-Almenara, Francesc Rubi-Carnacea, Eloy Opisso, Fran Valenzuela-Pascual.

**Investigation:** Maria Masbernat-Almenara.

**Methodology:** Maria Masbernat-Almenara, Eloy Opisso, Esther Duarte-Oller.

**Resources:** Francesc Rubi-Carnacea, Esther Duarte-Oller, Josep Medina-Casanovas.

**Software:** Fran Valenzuela-Pascual.

**Supervision:** Eloy Opisso, Esther Duarte-Oller, Josep Medina-Casanovas, Fran Valenzuela-Pascual.

**Validation:** Maria Masbernat-Almenara.

**Writing – original draft:** Maria Masbernat-Almenara, Francesc Rubi-Carnacea, Fran Valenzuela-Pascual.

**Writing – review & editing:** Maria Masbernat-Almenara, Francesc Rubi-Carnacea, Eloy Opisso, Esther Duarte-Oller, Josep Medina-Casanovas, Fran Valenzuela-Pascual.

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
