## [Decision Letter · Decision Letter 0]

13 Oct 2022

PONE-D-22-13676Developing An Assistive Technology Usability Questionnaire for People with Neurological DiseasesPLOS ONE

Dear Dr. Rubí-Carnacea,

Thank you for submitting your manuscript to PLOS ONE. After careful consideration, we feel that it has merit but does not fully meet PLOS ONE’s publication criteria as it currently stands. Therefore, we invite you to submit a revised version of the manuscript that addresses the points raised during the review process.

The reviewers agree on several important areas for revision, particularly concerning the presentation and justification for details about the methods.  The authors are advised to attend carefully to the recommendations in these areas.

We look forward to receiving your revised manuscript.

Kind regards,

Jeffrey Jutai

Academic Editor

PLOS ONE

Journal Requirements:

Reviewers' comments:

Reviewer's Responses to Questions

**Comments to the Author**

1. Is the manuscript technically sound, and do the data support the conclusions?

Reviewer #1: Partly

Reviewer #2: Yes

2. Has the statistical analysis been performed appropriately and rigorously? 

Reviewer #1: I Don't Know

Reviewer #2: Yes

3. Have the authors made all data underlying the findings in their manuscript fully available?

Reviewer #1: Yes

Reviewer #2: Yes

4. Is the manuscript presented in an intelligible fashion and written in standard English?

Reviewer #1: No

Reviewer #2: No

5. Review Comments to the Author

Reviewer #1: This article has some potential to be publish.

The manuscript, however, suffers from major errors, that authors can improve.

I outlined some (conceptual and terminology) and pertinence/need.

I suggested to make some changes/clarifications.

Please find the revised manuscript as an attachment.

Reviewer #2: Dear authors,

The aim of the project is Assistive Technology Usability Questionnaire for people with Neurological Diseases. The authors have included Delphi and different psychometric tests to analyze the data. Many parts are clear but there are some parts that need clarification. They are mainly in the method section, and you can see my comments and suggestions in the manuscript. For example, recruitment of end users are not written in the section. Perhaps a separate section “participants” that group all the information regarding professionals and end users will help the readers to understand who they are.

Another important point is Rasch model. Rasch model can do different analyses but the authors have chosen only to present the MnSq. I have not asked you to add more analyses here because your manuscript has already a number of analyses.

Please make the changes accordingly and consider my suggestions if appropriate.

6. PLOS authors have the option to publish the peer review history of their article (what does this mean?). If published, this will include your full peer review and any attached files.

Reviewer #1: **Yes: **Anabela Correia Martins

Reviewer #2: **Yes: **Helen Lindner

---

## [Author Response · Author response to Decision Letter 0]

16 Nov 2022

REVIEWER 1

1. Assistive Technologies, instead of Nervous System Diseases????

We thank the reviewer's observation. MESH term of Assistive Technologies is Self-Help Devices.

2. functioning, instead

We thank the reviewer's observation. However, we want to express functionality instead of functioning. 

3. This definition is very questionable! The authors have taken and written ipsis verbis from [10] who, in turn, took it from Fed Regist. 1991;56(60):4121... 1991!!!!!!!!!!!!!!!!! It will be useful to reformulate this text and, consequently, the conceptual framework of this definition.

According to the reviewer, we have added a new definition. Please see page 4, line 76 and below.

According to the WHO, assistive technology is a general term covering the systems and services related to delivering assistive products and services.[12] Assistive products aim to maintain or improve an individual's functioning and independence, promoting their well-being.[12]

4. To use the term "person with neurological diseases" is not entirely correct. I think the authors want to develop a tool to assess the usability of assistive technology products and user satisfaction and make it accessible to everyone, including those with communication limitations/restritions, cognitive/mental deficits/etc. It would not be appropriate to talk about "neurological disorders" in general, as there are some persons with neurological diseases, but who do not present any difficulty in understanding and answering the existing questionnaires.

We appreciate the reviewer's comment and agree; however, this questionnaire was developed and tested by people with neurological diseases. It is right that some people do not present mental/cognitive disorders, and the questionnaire will be easier for them. 

5. Why did you use the "Short Portable Mental Status Questionnaire for the Assessment of Organic Brain Deficit in Elderly Patients", since it is an instrument for the elderly population and the sample used ranged from 16 to 72 years (mean age 48 years)?

Thank you for your valuable observation. We agree with the reviewer's comment, and we chose the Pfeiffer test instead of MMSE because it is shorter, and we thought that should be too many questionnaires at the same time (Pfeiffer, Quest 2.0, and the new questionnaire). 

6.Don't you think it is too short??!! Aren't the users still learning how to dela with the assistive device?

We thank the reviewer for this suggestion. We chose at least one month because we performed the study in an acute and subacute hospital. People usually stay in the hospital for between 3-6 months, depending on the severity of the pathology and if they are in a private or public insurance. When the people are autonomous, they are discharged. We choose only one month to perform the pre and the post before being discharged. 

7. Explanation?

We appreciate the reviewer's comment. These patients did not use the AT at the moment of the interview; however, they used it in the past (last weeks, months)

We have added a sentence to clarify it. Please see table 1, pages 10 and 11, and below: 

cNothing means that he/she did not use assistive technologies at the moment of the interview; however, they used them in the past.

8. Why from the acute?!

We appreciate the editor's suggestion. The hospitals where we performed the study were the Institut Guttmann (Badalona) and the Hospital de l'Esperança (Barcelona) in Spain; both specialized in acute and subacute patients. 

9. Do you search for references/support? Please use a Reference.

Following the reviewer's suggestion, we have added a new reference. Please see page 14, line 270.

The statements of the questions are in first person to facilitate users' answers and usability experiences of the product subjectively [44].

First-person surveys in User Research | by Nikki Anderson | UX Collective. [cited 27 Oct 2022]. Available: https://uxdesign.cc/first-person-surveys-in-user-research-8732c5d9ab96

10. According to what?

We thank the reviewer's comment. We have clarified this information by adding a new sentence according to the comment. Please see page 14, line 272, and below

Once the questionnaire was finished, ten users (see demographic characteristics in table 5) read and answered the questionnaire to know if all the requirements from the Delphi study were met.

11. It was a criterium for inclusion... Not here...

Following the reviewer's advice, we have removed this part from the text. 

12. Initially the authors raised my expectation that they would develop a simple questionnaire to be used by people with neurological diseases and that there were no such resources adapted to you. How does this questionnaire differ from others? 

We agree with the reviewer's comment. This questionnaire has been created from the responses of both health professionals specialized in neurology and patients with neurological disorders, so it has been designed based on a user-centered design, unlike the questionnaires that currently exist for the same purpose. In addition, the existing questionnaires, as I have already mentioned in the article, are either very long or include questions/items that are not relevant for all countries, as is the case with QUEST, which includes a part of after-sales service and technical issues that many people with diseases of neurological origin would not know how to respond. In addition, our questionnaire has included the relevant items for these people and why they decide to use or not use the AT product.

13.How does it differ from the health conditions that led to the other assistive technology tools?

We appreciate the reviewer's comment. In the sentence, we would like to state that in the study, we analyze different tools which have been analyzed by different people with different neurological conditions. Moreover, for these reasons, the sample has good heterogeneity. 

REVIEWER 2

1.Can the authors elaborate a bit more the relationship between AT selection and outcome assessment?

Following your advice, we have added some sentences addressing your question. Please see page 4, line 81, and below:

For example, professionals prescribe assistive products without considering the user's needs. However, this clinical outcome assessment is important in clinical practice and research because it improves evidence and provides considerable feedback to healthcare professionals and patients, enhancing their empowerment with their opinions and needs and improving their quality of life[16]. 

2. Do you mean "usability of AT?". I would suggest you to add the word "usability" in this sentence. 

When I read any paper on usability, I always would like to see a definition of usability. if you have space, please add a definition of usability so the readers would know "what usability" are you focusing on since you mentioned the QUEST's technical items are difficult so your questionnaire does not focus on technical part.

We appreciate the reviewer's comment. We have added the Usability term and its definition. Please see page 3, line 68 and below:

According to the ISO 9241-11 [9], usability is the extent to which specified users can use a product to achieve specified goals with effectiveness, efficiency, and satisfaction in a specified context of use. Understanding this, the design of a product…

 3. I got a feeling some words are missing here. Please rephrase the sentence

We appreciate the reviewer's comment. We have rewritten the sentence. Please see page 4, line 91, and below:

There are other questionnaires developed exclusively to assess wheelchairs[22], and only a few are explicitly developed for assistive technologies. For example, the Quebec User Evaluation of Satisfaction with Assistive Technology (QUEST 2.0) [23] contains technical items such as weight, product dimensions, service delivery, repairs, and device services…

4. Can the authors clarify what you meant by limitations in the accessibility?

Following the reviewer's advice, we have modified the sentence. Please see page 5, lines 95 and below:

People with neurological diseases have limitations in the accessibility of the existing usability questionnaires, for example, in the comprehension of the questions or the answer form.

5. The recruitment of end users is an important part in this study. Now it is missing in the method. Please add in the recruitment process of end users.

We appreciate the reviewer's comment and have added a new sentence with this information. Please see page 5, line 114, and below:

Also, all the patients from the Institut Guttmann who voluntarily agreed to participate were interviewed to answer the questionnaire. 

6. CAn you the authors elaborate what these values are or what they represent.

Following the reviewer's suggestion, we have added a new sentence. Please see page 7, line 141, and below:

Subsequently, the users (people with neurological diseases) read the questionnaire to assess their understanding of the questions and if their answers could be represented with the proposed response values (numbers, traffic light colors, and faces).

7. I do not understand this sentence and how this sentence is related to the sample. Do the participants have to pass the SPMSQ test in order to be eligible to fill in the questionnaire? Please rephrase the sentence.

Following the reviewer's suggestion, we have added a new sentence. Please see page 7, line 148, and below:

Exclusion criteria moderate to severe cognitive impairment based on the Pfeiffer SPMSQ test [32] translated into Spanish[33].

8. since you only use ref 33, it is more appropriate to mention the authors' last names and not just several authors.

We thank the reviewer's observation; we have corrected it and added the missing reference. Please see page 8, line 165, and below

According to different authors,[35,36] an interval between two days and two weeks between the test-retest interviews is recommended.

9. Please add which ICC model you have used, such as ICC 2.1 , ICC 3.1 etc. Please add it here and also in the results

We thank the reviewer's remark. According to this remark, we have added a new sentence. Please see page 8, line 172, and below. 

Intraclass correlation coefficient (ICC) model 1.1 with a 95% of confidence interval (CI) was measured.

10. Please elaborate to your readers what do you mean by behaviour of items in the context of validity. 

Following the reviewer's suggestion, we have added a new sentence. Please see page 9, line 190, and below:

In order to further analyze the construct validity of the different items, the Rasch model [41] was carried out.

 11. this part of the analysis is an analysis of concurrent validity. Construct validity is an umbrella term for different validities. 

My suggestion is to change this term to "concurrent validity".

Thank you for your valuable observation. Following the reviewer's suggestion, we have modified the term. Please see page 8, line 181, and below: 

Concurrent validity. To know the relationship between the new questionnaire and another related questionnaire, QUEST 2.0[23] was chosen 

12 . which model are you using? rating scale model?

We thank the reviewer's observation. We are using the Rasch model because it is commonly used in the analysis of the construct validity of an instrument. Please see the reference:

Boone WJ. Rasch Analysis for Instrument Development: Why, When, and How? CBE Life Sci Educ. 2016;15. doi:10.1187/CBE.16-04-0148

13. Move this sentence to the previous parapraph

Following the reviewer's suggestion, we have moved the sentence to the previous paragraph. Please see page 9, line 196 and below: 

The study was developed following the COSMIN guidelines[42].

Ethical Approval

14. even though it is only 2 persons, it is not valid to ask end users that do not have any AT to answer the questions in the Delphi study, such as "would you change anything about our existing products?". 

We thank the reviewer's observation. These patients didn't use the AT at the moment of the interview; however, they have used it in the past (last weeks, months)

We will add a sentence to clarify it. Please see table 1 on page 11 and below: 

cNothing means that he/she did not use assistive technologies at the moment of the interview; however, they used it in the past

15.Did these experts involve in the rounds? Please clarify.

 Following the reviewer's advice, we have added a new sentence. Please see page 12, line 232, and below

Thirty-four experts from the Delphi study from the private neurorehabilitation hospital of acute neurological patients and the public hospital of acute stroke patients were selected to evaluate the items obtained from round two. 

16. Ergonomics is related to body posture and positions, often in workplace. My daily live needs is much bigger than body posture. Please explain why this item is categorized as ergonomics?

We thank the reviewer for this suggestion. According to the ISO 9241-11:2018(en), Ergonomics of human-system interaction — Part 11: Usability: Definitions and concepts, ergonomics/human factor is a scientific discipline concerned with the understanding of interactions among humans and other elements of a system, and the profession that applies theory, principles, data, and methods to design in order to optimize human well-being and overall system performance, 

Another definition: Ergonomics is the design and engineering of human-machine systems to enhance human performance.

Dempsey, Patrick G., Wogalter, Michael S., & Hancock, Peter A. (2000). What is in a name? Using terms from definitions to examine the fundamental foundation of human factors and ergonomics science. Theoretical Issues in Ergonomics Science, 1(1), 3-10.

Also, see the text below from Ahasan, R., Campbell, D., Salmoni, A., & Lewko, J. (2001). HFs/ergonomics of assistive technology. Journal of physiological anthropology and applied human science, 20(3), 187–197. https://doi.org/10.2114/jpa.20.187. 

How ergonomic designs could be helpful for special needs, Coleman and Pullinger (1993) noted some aspects of improvement features of disability and the aging process. Ergonomics application is important in the design and development of AT towards meeting the needs of disabled people. Hence by addressing the actual needs of these people, ergonomics principles are to be followed to enhance the mobility and communication of disabled people. Ergonomically designed AT usually enhance functional requirements that correctly fit the future application of the users meeting the needs of disabled people.

For these reasons, we thought that ergonomics fit in this category. 

17. Please provide information about these ten users? were they in the Delphi rounds? do you do AT? what are their diagnoses?

We appreciate the reviewer's comment. We have added a new sentence in order to clarify it. Please see page 14, line 271.

Once the questionnaire was finished, ten users (see demographic characteristics in table 5)

Also, we have added a new table with the demographic data of the ten users. Please see page 15, line 274, and below: 

Table 5. Demographic characteristics users from the questionnaire design. 

Questionnaire analysis n= 10

Demographic Characteristics Number Percentage

Age 

Between 18 and 25 years n=1 10%

Between 26 and 35 years n=2 20%

Between 36 and 45 years n=2 20%

Between 46 and 65 years n= 3 30%

More than 65 years n= 2 20%

Sex 

Female n=6 60%

Male n=4 40%

Pathology 

Spinal Cord Injury n=4 40%

Traumatic Brain Injury n= 2 20%

Stroke n= 3 30%

Other n= 1 10%

Assistive Technologies 

Wheelchair n= 8 80%

Canes n= 1 10%

Orthesis n=4 40%

Other n=1 10%

Nothinga n=1 10%

aNothing means that he/she did not use assistive technologies at the moment of the interview; however, they used it in the past

18. What is a usability test? please clarify

We thank the reviewer's observation, and we have added a new sentence in the introduction to clarify it. Please see page 4, line 73 and below:

For that reason, it is essential to perform a usability test during product development to know if it fits the user's needs. Usability testing refers to evaluating a product or service by testing it with representative users [11].

19. did these 51 patients not the end users in the Delphi study? if not, please replace the word "round" with " adminstration" or " occasion".

We agree with the reviewer's comment. We have changed the word "round" to "administration" to distinguish. Also, we expressed that the patients are different from the Delphi study. Please see page 16, line 285, and below:

A total of 51 people with neurological diseases consecutive recruited from the Institut Guttmann Hospital voluntarily agreed to participate in the study. These people were different from the Delphi study, and their demographic characteristics are summarized in Table 6. 

20. Brain injury is often acquired or traumatic. What are the brain injuries in these 4 patients?

We thank the reviewer's observation, and we have corrected it. Please see table 6 on page 16 and below: 

Table 6. Demographic characteristics of the 51 participants in the reliability and validity of NATU Quest. 

Demographic Characteristics Mean /Number Percentage

Age, range 16-72 years Mean = 48 

Sex 

Female n=20 39%

Male n= 31 61%

Pathology 

Spinal cord Injury n=22 43%

Traumatic Brain Injury n=3 6%

Stroke n=16 31%

Acquired Brain injury n=4 8%

Other neurological diseases n=6 12%

Education 

Secondary n=13 25%

High School Diploma n=1 2%

Certificate of Professional Standards n=15 29%

Higher Level Education Cycle n=7 14%

University Degree n=15 30%

Questionnaire form 

Self-administration n=24 53%

Interview n=27 47%

Assistive Technologies 

Wheelchair n=29 

Walking stick n=8 

Walker n=3 

Splint n=7 

Treadmill n=5 

21.Cognitive impairments are common in persons with neurological diseases. Do you have a criterion that the end users do not have cognitive impairments? 

And Pfeiffer test was not mentioned at all in the method. Please add this test in the method section.

We appreciate the reviewer's comment. Pfeiffer test was mentioned in the inclusion/exclusion criteria. Please see page 7, line 148, and below: 

Exclusion criteria moderate to severe cognitive impairment based on the Pfeiffer SPMSQ test [32] translated into Spanish[33].

22.a question about the expertise of doctors and PTs in assistive technology. Do doctors work with AT ? it is often occupational therapists and physiotherapists who have knowledge. Please add more information about doctors' expertise in AT in your country.

We thank the reviewer's comment. We have exchanged medical doctors with occupational therapists to make it more understandable. However, in the center where we performed the study, there is a transdisciplinary team, and the prescription of the orthoses is agreed upon by medical doctors, PT, occupational therapists, and orthopaedists. Please see page 18, line 325 and below: 

The items included in the questionnaire, the format of the questionnaire, and the answer form were derived through two rounds of a Delphi study[27] based on the opinion of 69 experts (neurorehabilitation professionals, such as occupational therapists and physiotherapists) and 63 users (people with neurological diseases). 

23. do you know if the other regions in Spain have similar experiences?

Following the reviewer's suggestion, we have added a new sentence. Please see page 19, line 343, and below: 

(1) Selection bias of the participants since most of them were from the same province. However, other regions have the same experiences. [45] 

24. no information on Pfeiffer test in the study.

We appreciate the reviewer's comment. Pfeiffer test was mentioned in the inclusion/exclusion criteria. Please see page 7, line 148, and below: 

Exclusion criteria moderate to severe cognitive impairment based on the Pfeiffer SPMSQ test [32] translated into Spanish[33].

25. the ATs in your study are mobility AT. Do you want to measure only mobility AT or other AT that support cognition? if yes, please add more information.

We thank the reviewer's observation. We only wanted to measure mobility AT. Others, such as cognition or audition, have other parameters and items.

26. I don't see if your test-retest reliability has compared with Quest 2. Please clarify.

We thank the reviewer's observation. Please see the results from pages 15 to 17 and in table 7 and below: 

Reliability and Validity

Sample description 

A total of 51 people with neurological diseases consecutive recruited from the Institut Guttmann Hospital voluntarily agreed to participate in the study. These people were different from the Delphi study, and their demographic characteristics are summarized in Table 6. Fifty-three percent of the participants answered the questionnaire through an interview due to physical limitations, and the rest (47%) answered it by themselves. First, the participants answered the new questionnaire and the QUEST 2.0. On average, the participants completed the new questionnaire within 102.40 seconds in the first administration and within 82.08 seconds in the second administration. The QUEST 2.0 was administered once before the NATU Quest, and the participants needed an average of 74 seconds to complete it. Participants scored an assistive product they had used in the last three months. All the patients answered all the items. 

Table 6. Demographic characteristics of the 51 participants in the reliability and validity of NATU Quest. 

Demographic Characteristics Mean /Number Percentage

Age, range 16-72 years Mean = 48 

Sex 

Female n=20 39%

Male n= 31 61%

Pathology 

Spinal cord Injury n=22 43%

Traumatic Brain Injury n=3 6%

Stroke n=16 31%

Acquired Brain injury n=4 8%

Other neurological diseases n=6 12%

Education 

Secondary n=13 25%

High School Diploma n=1 2%

Certificate of Professional Standards n=15 29%

Higher Level Education Cycle n=7 14%

University Degree n=15 30%

Questionnaire form 

Self-administration n=24 53%

Interview n=27 47%

Assistive Technologies 

Wheelchair n=29 

Walking stick n=8 

Walker n=3 

Splint n=7 

Treadmill n=5 

Reliability Results

The internal consistency reliability of the NATU Quest was analyzed using the Cronbach's Alpha[34] (α=0.895). This result can be interpreted as good reliability. 

Reliability through test-retest. A retest was performed 15 days after answering the two questionnaires for the first time to assess the reliability of the NATU Quest. Table 7 shows the weighted quadratic Kappa coefficient and Spearman's coefficient results of the NATU Quest. The results showed a moderate to considerable concordance between NATU Quest items test-retest in the Kappa coefficient because all the results were above 0,50. The results also showed a strong association between test-retest in the Spearman's coefficient (ρ = 0.818), significant with p-value < 0.0001. The results of the ICC showed good reliability (ICC = 0.869; CI 95% 0.781 to 0.923).

Table 7. NATU Quest reliability of the test-retest.

NATU Quest

Items Weighted Kappa 95% Confidence Interval

 Lower Limit Upper Limit

1 Effectiveness 0.727 0.535 0.918

2 Comfortable 0.733 0.645 0.821

3 Adaptability 0.585 0.434 0.735

4 Easy to put on/off 0.631 0.399 0.863

5 Safe 0.551 0.206 0.895

6 Functionality 0.685 0.541 0.828

7 Ergonomics 0.673 0.507 0.840

8 Easy to use 0.544 0.332 0.755

9 Easy to remember how to use it 0.624 0.298 0.950

10 Satisfaction 0.714 0.557 0.871

Spearman 0.818a 

aThe correlation is significant p< 0.0001 (bilateral).

Concurrent validity 

The correlation of the total scores between NATU Quest and QUEST 2.0 analyzed with the Spearman's coefficient was strong with ρ = 0.756 significant, with p-value < 0.0001

---

## [Decision Letter · Decision Letter 1]

13 Dec 2022

PONE-D-22-13676R1Developing An Assistive Technology Usability Questionnaire for People with Neurological DiseasesPLOS ONE

Dear Dr. Rubí-Carnacea,

Thank you for submitting your manuscript to PLOS ONE. After careful consideration, we feel that it has merit but does not fully meet PLOS ONE’s publication criteria as it currently stands. Therefore, we invite you to submit a revised version of the manuscript that addresses the points raised during the review process.

The reviewers have recommended minor revisions.  Please be sure to address all of their recommendations in your response to the reviews.==============================

We look forward to receiving your revised manuscript.

Kind regards,

Jeffrey Jutai

Academic Editor

PLOS ONE

Journal Requirements:

Additional Editor Comments (if provided):

The reviewers have recommended minor revisions. Please be sure to address all of their recommendations in your response to the reviews.

Reviewers' comments:

Reviewer's Responses to Questions

**Comments to the Author**

1. If the authors have adequately addressed your comments raised in a previous round of review and you feel that this manuscript is now acceptable for publication, you may indicate that here to bypass the “Comments to the Author” section, enter your conflict of interest statement in the “Confidential to Editor” section, and submit your "Accept" recommendation.

Reviewer #1: (No Response)

Reviewer #2: All comments have been addressed

2. Is the manuscript technically sound, and do the data support the conclusions?

Reviewer #1: (No Response)

Reviewer #2: Yes

3. Has the statistical analysis been performed appropriately and rigorously? 

Reviewer #1: (No Response)

Reviewer #2: Yes

4. Have the authors made all data underlying the findings in their manuscript fully available?

Reviewer #1: (No Response)

Reviewer #2: (No Response)

5. Is the manuscript presented in an intelligible fashion and written in standard English?

Reviewer #1: (No Response)

Reviewer #2: Yes

6. Review Comments to the Author

Reviewer #1: Most of the comments from the 1st revision were not answered, including terminology that I corrected, as well as all requests for justification of some methodological options and interpretation of the results. The minor changes that were made were not highlighted in the text, which made my analysis difficult.

Reviewer #2: Dear authors,

The manuscript has improved a lot. I have several minor suggestions that I would like the authors to consider them.

7. PLOS authors have the option to publish the peer review history of their article (what does this mean?). If published, this will include your full peer review and any attached files.

Reviewer #1: No

Reviewer #2: **Yes: **Helen Lindner

---

## [Author Response · Author response to Decision Letter 1]

27 Dec 2022

Dear Editor of PLOS ONE, 

I am pleased to resubmit the revised version of PONE-D-22-13676 " Developing An Assistive Technology Usability Questionnaire for People with Neurological Diseases" for your consideration. We are very grateful for the excellent suggestions and comments from the reviewers. As you suggested, we have carefully considered all the suggestions and comments and made revisions accordingly. Thanks again for those valuable suggestions, which have helped us further improve the quality and clarity of this manuscript. We have addressed each reviewer's requirements as outlined below, which are indicated in the manuscript in yellow.

REVIEWER 1

In reference to the comments to reviewer 1, we apologize if he/she thinks that we did not answer the questions. Please see the answers below: 

1. Assistive Technologies, instead of Nervous System Diseases????

We thank the reviewer's observation. We have added “Assistive technologies” instead of “Nervous System Diseases”. Please see page 2, line 47, and below: 

“Keywords: Assistive technologies, Neurological Rehabilitation, Questionnaire Design, Self-Help Devices, User-centered Design”

2. functioning, instead

Thank you for the comment. According to the comment, we have changed the word functionality to functioning. Please see page 2, line 59 and below:

“improve their functioning and autonomy in their daily life.”

Also, see page 10 line 211 and below: 

 

“Round two involved 59 people (27 experts and 32 users) (Table 1). This round obtained 15 items and data about the scale. The items were: "effectiveness", "comfort", "adaptability", "easy to put on/off", "safe", "lightweight", "functioning", "ergonomic", "economical", "affordable", "easy to use", "feedback", "stimulating," "monitored" and "movement facilitator.”

Also see page 18 line 331 and below: 

“The item "functioning" appear in the USAT-WM.[22]”

3. This definition is very questionable! The authors have taken and written ipsis verbis from [10] who, in turn, took it from Fed Regist. 1991;56(60):4121... 1991!!!!!!!!!!!!!!!!! It will be useful to reformulate this text and, consequently, the conceptual framework of this definition.

Thank you very much for your comment. The definition was too old, so we have added a new definition. Please see page 4, line 76 and below.

“According to the WHO, assistive technology is a general term covering the systems and services related to delivering assistive products and services.[12] Assistive products aim to maintain or improve an individual's functioning and independence, promoting their well-being.[12]”

4. To use the term "person with neurological diseases" is not entirely correct. I think the authors want to develop a tool to assess the usability of assistive technology products and user satisfaction and make it accessible to everyone, including those with communication limitations/restritions, cognitive/mental deficits/etc. It would not be appropriate to talk about "neurological disorders" in general, as there are some persons with neurological diseases, but who do not present any difficulty in understanding and answering the existing questionnaires.

We appreciate the reviewer's comment; however, this questionnaire has been developed for people with neurological diseases. The authors do not aim to develop a tool to assess the usability of assistive technology for other patients´ typologies. 

5. Why did you use the "Short Portable Mental Status Questionnaire for the Assessment of Organic Brain Deficit in Elderly Patients", since it is an instrument for the elderly population and the sample used ranged from 16 to 72 years (mean age 48 years)?

Thank you for your valuable observation. We agree with the reviewer's comment. We chose the Pfeiffer test instead of MMSE because it is shorter and more manageable for these patients. We included this point in the limitations section. Please see page 19, line 344, and below: 

“2) Although all the users who participated in the validation had a neurological disease, the authors chose the Pfeiffer SPMSQ to assess the cognitive problems because it is short and quick to answer. However, Pfeiffer SPMSQ does not accurately assess all possible cognitive deficits, and it is not sensitive enough to detect low or mild cognitive deficits.”

6.Don't you think it is too short??!! Aren't the users still learning how to dela with the assistive device?

We appreciate the comment. We chose at least one month for practical reasons because we performed the study in a hospital where most patients are acute and subacute. In addition, other usability questionnaires, such as PIADS and QUEST, do not express this point. 

Moreover, we have added this to the limitations section. Please see page 19, line 351, and below: 

“(4) For practical reasons we chose that the users have used the product at least for one month, however, probably is not enough time to test a product.”

7. Explanation?

We appreciate the reviewer's comment. These patients did not use the AT at the moment of the interview; however, they used it in the past (last weeks, months)

Following the reviewer's comment, we have added a sentence to clarify it. Please see table 1, pages 10 and 11, and table 5, page 15 and below: 

“Nothing means that he/she did not use assistive technologies at the moment of the interview; however, she/he used them in the past.”

8. Why from the acute?

We appreciate the reviewer's comment. In the study, we included acute, subacute and chronic patients, however, in order to avoid misunderstandings we have deleted hospitals’ characteristics.

Please see page 5, line 112 and below: 

“A google form questionnaire was sent via email to the neurological healthcare professionals from the Institut Guttmann, Spain and Hospital de l’Esperança, Spain.

Please see page 11, line 230 and below: 

“Thirty-four experts from the Delphi study from the Institut Guttmann and the Hospital de l'Esperança were selected to evaluate the items obtained from round two..”

9. Read the questionnaire to assess their understanding of the questions and if their answers could be represented with the proposed response.

Thank you for your appreciation. We have added a new sentence in order to clarify the sentence. Please see page 6, line 137 and below:

“Subsequently, ten users (people with neurological diseases) read the questionnaire to assess their understanding of the questions and if their answers from the Delphi study could be represented with the proposed response values (numbers, traffic light colors, and faces).”

10. Do you search for references/support? Please use a Reference.

Thank you for your observation. Following your advice, we have added a new reference. Please see page 14, line 268, and below: 

“The statements of the questions are in first person to facilitate users' answers and usability experiences of the product subjectively [44].”

First-person surveys in User Research | by Nikki Anderson | UX Collective. [cited 27 Oct 2022]. Available: https://uxdesign.cc/first-person-surveys-in-user-research-8732c5d9ab96

11. According to what?

We thank the reviewer's comment. We have clarified this information by adding a new sentence according to the comment. Please see page 14, line 268, and below:

“Once the questionnaire was finished, ten users (see demographic characteristics in table 5) read and answered the questionnaire to know if all the requirements from the Delphi study were met.”

12. It was a criterium for inclusion... Not here...

Thank you for your comment. Following the reviewer's advice, we have removed this part from the text. 

13. Initially the authors raised my expectation that they would develop a simple questionnaire to be used by people with neurological diseases and that there were no such resources adapted to you. How does this questionnaire differ from others? 

Thank you for your comment. As we have already mentioned in the article, the existing questionnaires are either very long or include questions/items that are not relevant for all countries, as is the case with QUEST, which includes a part of after-sales service and technical issues that many people with diseases of neurological origin would not know how to respond. In addition, our questionnaire has included the relevant items for these people and why they decide to use or not use the AT product.

14.How does it differ from the health conditions that led to the other assistive technology tools?

We appreciate the reviewer's comment. The questionnaire was designed by people with neurological diseases following their particular needs. As we mentioned in future work, validating this questionnaire with a different population would be interesting.

REVIEWER 2

1.I agree with your two references and my suggestion is to change item 7: ________adapts to my special needs. My daily live needs is broader than my special needs. 

As your reference said "....for special needs" and this instrument is about AT for persons with neurological illnesses,

We agree with the reviewer's comment. Following the reviewer's suggestion, we have changed item 7. Please see pages 13 and 14, table 4, and below: 

Table 4. Questions and the items that are involved in each question. 

Questions Items

1 I believe that ¬_____ can help me improve my functional independence Effectiveness

2 I feel comfortable wearing/using ____ Comfortable

3 _____ adapts to my characteristics and needs Adaptability

4 Donning/Doffing …………….. is quick and easy for me. Easy to put on / off

5 I feel safe using/wearing _____ / _____ is safe in its use Safe

6 _____ allows me to achieve my goal/ allows me to perform a movement/action that I could not do before Functionality/ movement facilitator

7 _____ adapts to my special needs. Ergonomics

8 In general, _____ is easy to use Easy to use

9 Information and instructions of use______ are easy to understand and easy to remember Easy to remember how to use it

10 Overall, I am satisfied with______ Satisfaction

2. what does SPMSQ stand for?

We have clarified this information by adding a new sentence. Please see page 7, line 146, and below: 

“Exclusion criteria moderate to severe cognitive impairment based on the Pfeiffer Short Portable Mental State Questionnaire (Pfeiffer SPMSQ).”

3.since it is only 1 person, do not use the word "they" but he/she

We appreciate the reviewer's observation, and we have corrected it. Please see table 1, page 11 and line 224; and table 5, page 15, line 273, and below: 

“Nothing means that he/she did not use assistive technologies at the moment of the interview; however, she/he used it in the past.”

-----

“In the second round, 36% did not answer the questionnaire. Others are, for example, orthopedic professionals and engineers. Nothing means that he/she did not use assistive technologies at the moment of the interview; however, she/he used them in the past. Users are different from round 1.”

In the name of the authors, I hope that this revised manuscript answers all the concerns contained in the reviews, and we are grateful for the thought and effort the reviewers have put into these reviews. 

Yours sincerely.

Francesc Rubí-Carnacea

---

## [Editor Report · Decision Letter 2]

18 Jan 2023

Developing An Assistive Technology Usability Questionnaire for People with Neurological Diseases

PONE-D-22-13676R2

Dear Dr. Rubí-Carnacea,

We’re pleased to inform you that your manuscript has been judged scientifically suitable for publication and will be formally accepted for publication once it meets all outstanding technical requirements.

Kind regards,

Jeffrey Jutai

Academic Editor

PLOS ONE

Additional Editor Comments (optional):

The authors have satisfactorily addressed the reviewers' concerns.
---

## [Editor Report · Acceptance letter]

23 Jan 2023

PONE-D-22-13676R2 

Developing An Assistive Technology Usability Questionnaire for People with Neurological Diseases 

Dear Dr. Rubí-Carnacea:

I'm pleased to inform you that your manuscript has been deemed suitable for publication in PLOS ONE. Congratulations! Your manuscript is now with our production department. 

Kind regards, 

on behalf of

Dr. Jeffrey Jutai 

Academic Editor

PLOS ONE